rsos.royalsocietypublishing.org

Subject Areas:
computer modelling and simulation/graph theory/applied mathematics

Keywords:
complex network, scale-free network, short-term information spreading strategy, spreading efficiency, initially informed vertices

Author for correspondence:
Yunfeng Deng
e-mail: 13910185162@139.com

# Improving short-term information spreading efficiency in scale-free networks by specifying top large-degree vertices as the initial spreaders

Shuangyan Wang[1], Yunfeng Deng[2] and Ying Li[1]

[1]School of Engineering and Technology, China University of Geosciences, Beijing, People's Republic of China
[2]Chinese Academy of Governance, Beijing, People's Republic of China

 SW, 0000-0001-7564-3674

The positive function of initially influential vertices could be exploited to improve spreading efficiency for short-term spreading in scale-free networks. However, the selection of initial spreaders depends on the specific scenes. The selection of initial spreaders needs to offer low complexity and low power consumption for short-term spreading. In this paper, we propose a selection strategy for efficiently spreading information by specifying a set of top large-degree vertices as the initially informed vertices. The essential idea behind the proposed selection strategy is to exploit the significant diffusion of the top large-degree vertices at the beginning of spreading. To evaluate the positive impact of initially influential vertices, we first build an information spreading model in the Barabási–Albert (BA) scale-free network; next, we design 54 comparative Monte Carlo experiments based on a benchmark strategy and the proposed selection strategy in different BA scale-free network structures. Experimental results indicate that (i) the proposed selection strategy can significantly improve spreading efficiency in the short-term spreading and (ii) both network size and number of hubs have a strong impact on spreading efficiency, while the number of initially informed vertices has a weak impact. The proposed selection strategy can be employed in short-term spreading, such as sending warnings or crisis information spreading or information spreading in emergency training or realistic emergency scenes.

rsos.royalsocietypublishing.org R. Soc. open sci. 5: 181137

# 1. Introduction

In the analysis of social networks, a typical problem is how to effectively diffuse information by exploiting different spreading strategies and/or different selection methods in varied networks. Effective information spreading can assist in rapidly diffusing expected positive and important information. For example, efficient methods for spreading information positively can be employed in warning information spreading in emergency training or realistic emergency scenes. Considerable research has been performed to improve spreading efficiency, such as the shortest spreading paths [1–4], spreading strategies [5–8], influential spreader [9–14], spreading process [15–19] and spreading behaviours [20,21].

More specifically, Pei et al. [22,23] demonstrated that information could be diffused rapidly by preferentially spreading to the influential connected vertices. Yang et al. [24] revealed that infection can spread to a larger part of the network when small-degree vertices were selected more frequently as targets; however, spreading can be suppressed severely when a small set of hub vertices dominated the dynamics. Tao Zhou et al. [1] indicated that spreading can be maximized if the contact probability determined by a generic function of its degree $W(k)$ was chosen to be inversely proportional to the vertex degree, i.e. $W(k) \sim k^{-1}$. Yang et al. [25,26] revealed that the vertices with the larger k-shell were more influential spreaders in the spreading complex network; and they improved and proposed the relevant measuring methods based on the traditional K-Shell decomposition. Lei Gao et al. [27] indicated that the information can be diffused efficiently by selecting and preferentially spreading to small-degree neighbours with small informed density. Hyoungshick Kim et al. [28] studied the effective neighbours selection strategies for maximum information diffusion in online social networks; they indicated that an effective neighbours selection strategy was to use vertex degree information for short-term spreading. Thus, for information spreading in an emergency, selecting influential neighbours based on the vertex degree is an effective approach for maximally diffusing the information.

In addition to the methods described above, the selection of a set of specific vertices as the initially influenced concepts is also an effective method to efficiently diffuse information, which is because of the significant influence of those vertices on their neighbours or connected vertices [29,30]. For example, Elliot Anshelevich et al. [31] provided efficient heuristics for selecting a subset of the actors (the seed set) to initialize with information with the goal of maximizing the final set of actors who believe and act upon that information. These researchers posited that (i) it is necessary to determine initial vertices to inject with the information, and (ii) it is essential to use the social communication network to diffuse a warning. Lei Guo et al. [32] proposed an improved method to choose the initial spreaders according to the different distances between vertices, and they found that spreading influence was more significant when the distance between initial spreaders was close to the average distance of a network. Yu-Hsiang Fu et al. [33] proposed a two-step framework as a vertex ranking measure, and they indicated that the proposed method was capable of identifying the most influential vertices as initial spreaders that disseminate information in different networks. By contrast, Hyoungshick Kim et al. [28] thought that it was unacceptable to select a set of any arbitrary $k$ nodes as the initially influential vertices for effectively spreading information in online social networks. Information can always be diffused if the spreading time is adequate. However, we think that for short-term spreading with a limited spreading time, such as spreading in an emergency, it is probably acceptable to select a set of specific vertices as the initially influential vertices to spread information efficiently in social networks.

In this paper, we propose a selection strategy for improving short-term spreading efficiency, and we evaluate the impact of the selection strategies with different *Initially Informed Vertices* (i.e. the IIVs) on the efficiency of information spreading in different network structures.

(1) We first build an information spreading model based on the multi-agent modelling method.
(2) We design 54 groups of comparative experiments. (a) We employ two selection strategies in these comparative experiments, i.e. the proposed selection and a benchmark strategy. Because of the low complexity of degree centrality, we choose degree centrality as the metric to measure the influence of initial spreaders in the proposed selection strategy, and we select a set of top large-degree vertices as the initial spreaders in networks. (b) To evaluate the impact of different settings of the number of initially informed vertices, we design different sizes for initially informed vertices. (c) To evaluate the impact of different network structures, we design different network sizes and/or different numbers of hubs for Barabási–Albert (BA) scale-free networks. (d) To enhance the reliability of experiments, we conduct plentiful, repeated and random experiments by exploiting the Monte Carlo experiments.
(3) We evaluate spreading efficiency with different experimental configurations by comparing the *Cumulative Frequency Distribution* of spreading time, which are outputted by Monte Carlo experiments.

rsos.royalsocietypublishing.org R. Soc. open sci. **5**: 181137

The contribution of this work can be summarized as follows: (i) We propose a selection strategy for improving information spreading efficiency by specifying a set of top large-degree vertices as the initially informed vertices in short-term spreading; and (ii) We design a series of comparative experiments to evaluate the impact of the strategies with different initially informed vertices on information spreading efficiency in different network structures.

The rest of this paper is organized as follows: Section 2 provides a brief introduction to the problems that need to be addressed in this paper. Section 3 describes the proposed method, including the proposed information spreading model and the detailed experimental process. Sections 4 and 5 present and discuss the results of the experiments, respectively. Section 6 presents the study's conclusions.

# 2. Problem statement

There are two critical problems that need to be addressed to evaluate the impact of the selection strategies with different IIVs on spreading efficiency; and to employ a positive impact to enhance the spreading efficiency:

(1) Who should be specified to improve spreading efficiency in short-term spreading?

The different selection strategies with different IIVs could probably impact spreading efficiency, and we can improve spreading efficiency by exploiting the positive impact. A problem arises in this case: who (i.e. the IIVs) should be specified to improve spreading efficiency in short-term spreading?

The initially informed vertices could be determined according to a specific kind of metric, such as degree centrality, k-shell or betweenness. There are many metrics employed to measure the influence of vertices in the networks. We think that the degree centrality is the best choice for our study because (i) for large-scale networks or extremely complex networks, degree centrality generally is the simplest metric to be calculated that is suitable for employment in short-term spreading with a limited spreading time, and (ii) degree centrality can measure the direct diffusion of vertices and the large-degree vertices are connected with most vertices by only one edge. This characteristic is suitable to be employed in short-term spreading with a limited time. Therefore, we choose degree centrality as the measurement metric, and we specify a set of top large-degree vertices as the initially informed vertices in the proposed strategy (i.e. the SIIVs strategy).

(2) How many specific initial spreaders are sufficient to improve spreading efficiency?

The different settings of the number of IIVs could impact spreading efficiency. In this case, how many specific IIVs are sufficient to improve spreading efficiency?

To address the above question, we first design several groups of experiments by setting different numbers of IIVs; and then we evaluate the impact of different settings of the number of IIVs on spreading efficiency. More details will be presented in §3.

# 3. Material and methods

In this section, we will describe the details of the proposed efficient method for spreading information in scale-free networks, including (1) the basic ideas behind the proposed information spreading model, (2) the designed comparative experiments and (3) the Monte Carlo experiments employed in this paper.

## 3.1. Information spreading model

In this subsection, we give a brief introduction to the information spreading model, including the assumptions and the information spreading mechanism in this model. We build the information spreading model based on the multi-agent modelling, by exploiting the software Anylogic [34].

### 3.1.1. Model assumptions.

(i) *Assumption 1*: A number of vertices would be informed at the beginning of experiments. In the spreading model, we assume that a number of vertices would be informed at the beginning of experiments, and they are termed the initially informed vertices (i.e. the IIVs); they also are the initial information spreaders in the networks.

(ii) *Assumption 2*: Uninformed vertices become informed vertices after receiving information.
In our spreading model, to simplify the spreading process, we assume that the uninformed vertices will become the informed vertices immediately after receiving the information.

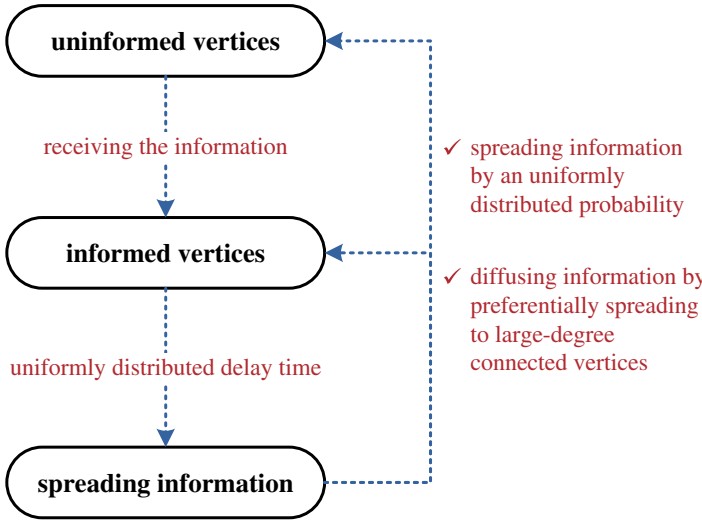

**Figure 1.** A simple flowchart of the information spreading mechanism.

(iii) *Assumption 3*: There is a delay period before the informed vertices spread the information.

We consider that the persons who know the information cannot spread the information immediately, and the waiting time before they spread the information varies for different people. Therefore, there is a delay period before the informed vertices spread the information; the delay time conforms to a uniform distribution of 1–5 s in our spreading model.

(iv) *Assumption 4*: Informed vertices spread the information based on different probabilities.

We consider that the spreading probabilities of persons who know the information are different. Thus, we assume the probability conforms to a uniform distribution of 0 to 1.

(v) *Assumption 5*: All informed vertices will diffuse the information.

To evaluate the spreading efficiency of the proposed method, we assume that all informed vertices will spread the information. According to *Assumption 3* and *Assumption 4*, we know that all informed vertices will spread information with different delay times and spreading probabilities.

(vi) *Assumption 6*: All informed vertices diffuse the information by preferentially spreading to those connected neighbouring vertices with large degrees.

It would take an extended period of time to converge in the spreading process when adopting a random spreading strategy. By contrast, the simulation time for spreading is generally satisfied when using a spreading strategy where all vertices diffuse the information by preferentially spreading to the influential connected vertices. Thus, in order to reduce the computational cost of simulating experiments, we assume that all informed vertices will diffuse information by preferentially spreading to the influential vertices [22,23], and we measure the influential vertices by exploiting degree centrality, i.e. all informed vertices diffuse the information by preferentially spreading to those connected neighbouring vertices with large degrees.

### 3.1.2. Information spreading mechanism.

Based on the above assumptions, we propose an information spreading mechanism for each vertex in the networks; a simple flowchart of the information spreading mechanism is in figure 1.

At the beginning of experiments, all vertices are initially set as uninformed vertices; meanwhile, a set of specific (or random) uninformed vertices are informed at first (i.e. the IIVs). The vertices that know the information then become the informed vertices, which have qualifications to diffuse information. After a random delay period, the informed vertices can diffuse information by preferentially spreading to large-degree connected vertices based on random probabilities.

## 3.2. Design of comparative experiments

In this subsection, we give a detailed introduction to the design of our comparative experiments, including the experimental configurations and the designed Monte Carlo experiments.

rsos.royalsocietypublishing.org    R. Soc. open sci. **5**: 181137

**Table 1.** The designed 18 comparative experiments in the networks with 500 people.

rsos.royalsocietypublishing.org    R. Soc. open sci. **5**: 181137

| **SIIVs**: *random initially informed vertices.* | | | | |
|---|---|---|---|---|
| **RIIVs**: *specific initially informed vertices.* | | | | |
| network size: 500 people | | | | |
| number of the hubs in scale-free networks with 500 people | 2 hubs (0.4% × network size) | number of RIIVs | 5% | 25 |
| | | | 10% | 50 |
| | | | 15% | 75 |
| | | number of SIIVs | 5% | 25 |
| | | | 10% | 50 |
| | | | 15% | 75 |
| | 3 hubs (0.6% × network size) | number of RIIVs | 5% | 25 |
| | | | 10% | 50 |
| | | | 15% | 75 |
| | | number of SIIVs | 5% | 25 |
| | | | 10% | 50 |
| | | | 15% | 75 |
| | 4 hubs (0.8% × network size) | number of RIIVs | 5% | 25 |
| | | | 10% | 50 |
| | | | 15% | 75 |
| | | number of SIIVs | 5% | 25 |
| | | | 10% | 50 |
| | | | 15% | 75 |

(i) *Configuration 1*: Two selection strategies are configured for selecting IIVs.

We configure two selection strategies. (a) *Specific* IIVs (i.e. the SIIVs) are selected. We choose degree centrality as the metric to measure the influence of initial spreaders, i.e. the initially informed vertices, and we specify a set of top large-degree vertices as the IIVs. (b) *Random* IIVs (i.e. the RIIVs) are selected. We randomly select a set of vertices as the IIVs in networks.

(ii) *Configuration 2*: Different numbers of initially informed vertices are set.

We configure different numbers of the IIVs in this paper and evaluate the impact of different sizes of the IIVs on spreading efficiency. It also should be noted that we compare spreading efficiency when using different selection strategies for spreading to the same number of IIVs in our comparative experiments.

We configure 3 numbers of initially informed vertices for each selection strategy in each network according to percentages based on the corresponding network size, i.e. the 5%, 10% and 15%. For example, there are 3 numbers of initially informed vertices for spreading with the proposed strategy in the network with 500 people and 2 hubs, i.e. 25, 50 and 75, respectively; see tables 1–3.

(iii) *Configuration 3*: Information spreading efficiency is evaluated in different scale-free networks.

It is well-known that scale-free networks exhibit many characteristics of realistic phenomena [35]. In this paper, we conduct our experiments based on the classical Barabási–Albert (BA) scale-free network [36]. It also should be noted that different network structures (i.e. the different scenes in real life probably) could affect the efficiency of the proposed method (i.e. the SIIVs strategy) on the spreading efficiency. Therefore, we design different BA scale-free networks by configuring different network sizes and/or the different numbers of hubs in BA scale-free networks.

As seen in figure 2, in this paper, we design 9 scale-free networks with different network sizes and different numbers of hubs. We also configure 3 network sizes, i.e. 500 people, 1000 people

**Table 2.** The designed 18 comparative experiments in the networks with 1000 people.

| | | | | |
|---|---|---|---|---|
| **SIIVs**: *random initially informed vertices.* | | | | |
| **RIIVs**: *specific initially informed vertices.* | | | | |
| network size: 1000 people | | | | |
| number of the hubs in scale-free networks with 1000 people | 4 hubs (0.4% × network size) | number of RIIVs | 5% | 50 |
| | | | 10% | 100 |
| | | | 15% | 150 |
| | | number of SIIVs | 5% | 50 |
| | | | 10% | 100 |
| | | | 15% | 150 |
| | 6 hubs (0.6% × network size) | number of RIIVs | 5% | 50 |
| | | | 10% | 100 |
| | | | 15% | 150 |
| | | number of SIIVs | 5% | 50 |
| | | | 10% | 100 |
| | | | 15% | 150 |
| | 8 hubs (0.8% × network size) | number of RIIVs | 5% | 50 |
| | | | 10% | 100 |
| | | | 15% | 150 |
| | | number of SIIVs | 5% | 50 |
| | | | 10% | 100 |
| | | | 15% | 150 |

rsos.royalsocietypublishing.org    R. Soc. open sci. **5**: 181137

and 2000 people. We configure the number of hubs in each network according to percentages based on the corresponding network size, i.e. 0.4%, 0.6% and 0.8%. For example, there are 3 configurations of the number of hubs for the network with 500 people, i.e., 2, 3 and 4, respectively; see tables 1–3.

(iv) *Configuration 4*: Experiments would terminate when all vertices have been informed or when the number of uninformed vertices becomes constant.

In short-term spreading, as with realistic emergency training or emergency scenes, the commander tends to know the number of vertices that know the information during a limited period. However, for the completeness of our experiments, we record the entire spreading process and the convergence time of the entire spreading process. Thus, we terminate the experiments when all vertices have been informed or the number of uninformed vertices is constant; and we record the simulation time as the convergence time of the entire spreading process.

We conduct experiments by exploiting the Monte Carlo experiment in Anylogic [34] to avoid the effect of randomness in the multi-agent information spreading model. Monte Carlo experiments are a broad class of computational algorithms that rely on repeated random sampling to obtain numerical results. The essential concept of these experiments is using randomness to solve problems that might be deterministic in principle [37].

There are 54 Monte Carlo experiments based on the above descriptions. We create a defined randomness generator to generate random numbers ranging from 0 to 1000 (excluding 1000). We design 1000 repeated simulations in each Monte Carlo experiment. The convergence time of spreading is configured as the objective parameter of the Monte Carlo experiments. A frequency statistic of 1000 convergence time can be generated in each Monte Carlo experiment. We set 0.1 as the interval of the frequency statistics, and we set 200 as the number of intervals in the frequency statistics.

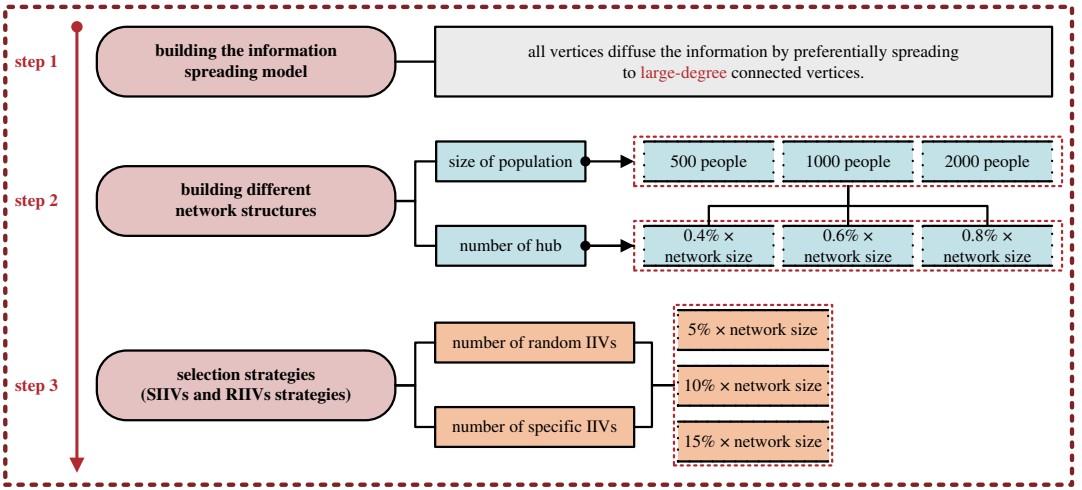

**Figure 2.** Process of the comparative experiments designed in this paper.

**Table 3.** The designed 18 comparative experiments in the networks with 2000 people.

| | | | | |
|---|---|---|---|---|
| **SIIVs**: *random initially informed vertices.* | | | | |
| **RIIVs**: *specific initially informed vertices.* | | | | |
| network size: 2000 people | | | | |
| number of the hubs in scale-free networks with 2000 people | 8 hubs (0.4% × network size) | number of RIIVs | 5% | 100 |
| | | | 10% | 200 |
| | | | 15% | 300 |
| | | number of SIIVs | 5% | 100 |
| | | | 10% | 200 |
| | | | 15% | 300 |
| | 12 hubs (0.6% × network size) | number of RIIVs | 5% | 100 |
| | | | 10% | 200 |
| | | | 15% | 300 |
| | | number of SIIVs | 5% | 100 |
| | | | 10% | 200 |
| | | | 15% | 300 |
| | 16 hubs (0.8% × network size) | number of RIIVs | 5% | 100 |
| | | | 10% | 200 |
| | | | 15% | 300 |
| | | number of SIIVs | 5% | 100 |
| | | | 10% | 200 |
| | | | 15% | 300 |

# 4. Results

## 4.1. Experimental environment and testing data

We conduct the designed experiments using the software Anylogic [34] on a powerful workstation computer. Moreover, details of the proposed information spreading model, the designed 54 comparative experiments, the Monte Carlo experiments and the relevant testing data introduced in §3. It needs to be

rsos.royalsocietypublishing.org    R. Soc. open sci. **5**: 181137

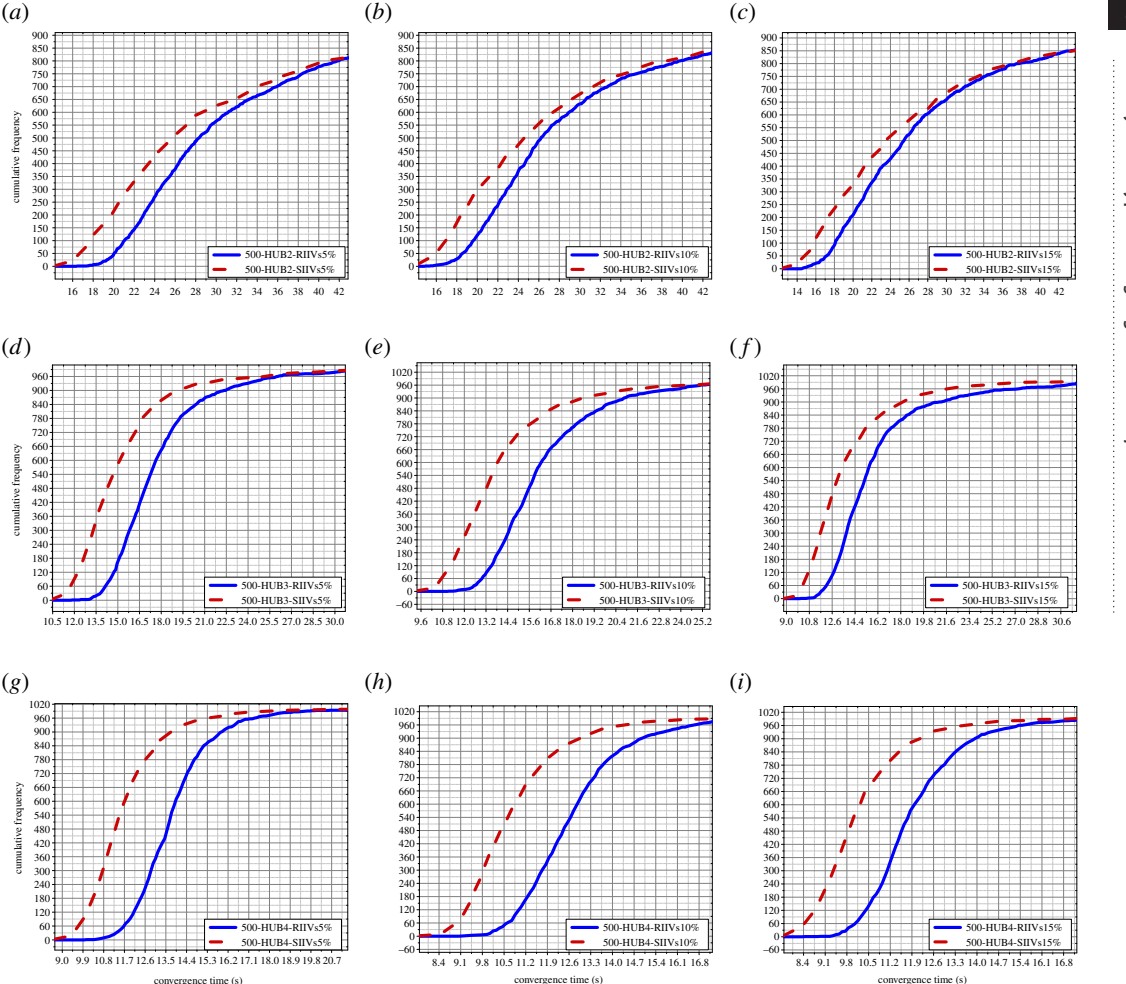

**Figure 3.** Results of the Monte Carlo experiments with 500 people. $(a) - (c)$ illustrate the results of Monte Carlo experiments with 500 people and 2 hubs; $(d) - (f)$ illustrate the results of Monte Carlo experiments with 500 people and 3 hubs; $(g) - (i)$ illustrate the results of Monte Carlo experiments with 500 people and 4 hubs. RIIVs: random initially informed vertices; SIIVs: specific initially informed vertices. 5%: the 5% of 500 initially informed vertices. 10%: the 10% of 500 initially informed vertices. 15%: the 15% of 500 initially informed vertices.

noted that we simplify the information spreading process, and the convergence time of experiments seems too small. However, the real convergence time is possibly dozens of times that of the results, and the experimental results are just used to reflect the comparison of the two strategies.

## 4.2. Results of the Monte Carlo experiments with 500 people

In this subsection, the results of the Monte Carlo experiments with 500 people are presented, including the results of Monte Carlo experiments with 500 people and 2 hubs, 3 hubs, and 4 hubs, respectively; see figure 3. The results indicate that for short-term spreading with limited time, the SIIVs strategy is capable of significantly improving spreading efficiency in networks with 500 people.

The improvement of spreading efficiency when using the proposed strategy is increasingly significant with the increase in the number of hubs. The improvement of spreading efficiency when using the proposed strategy is not increasingly significant with the increase in the number of IIVs, and there is a weakly decreased improvement.

## 4.3. Results of the Monte Carlo experiments with 1000 people

In this subsection, the results of Monte Carlo experiments with 1000 people are presented, including the results of Monte Carlo experiments with 1000 people and 4 hubs, 6 hubs and 8 hubs, respectively; see

rsos.royalsocietypublishing.org    R. Soc. open sci. **5**: 181137

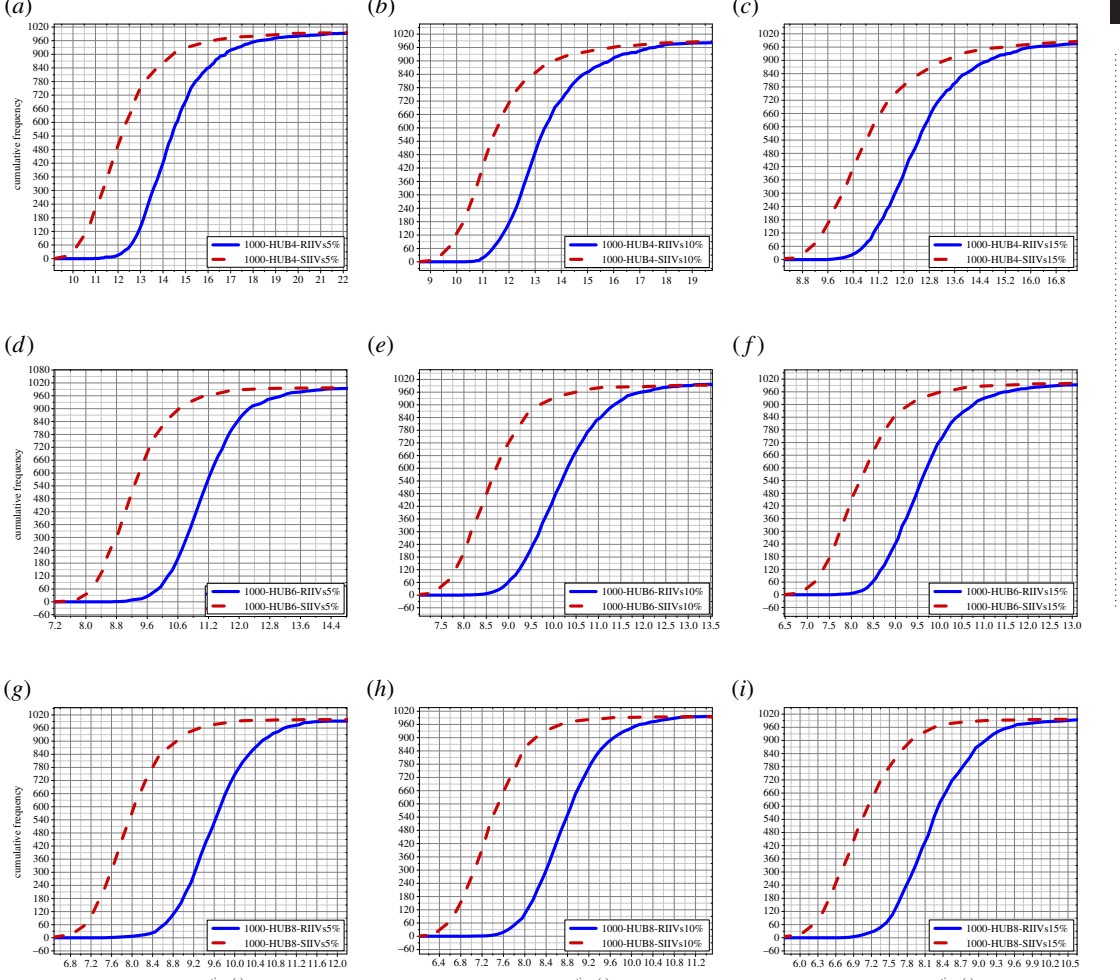

**Figure 4.** Results of the Monte Carlo experiments with 1000 people. $(a) - (c)$ illustrate the results of Monte Carlo experiments with 1000 people and 4 hubs; $(d) - (f)$ illustrate the results of Monte Carlo experiments with 1000 people and 6 hubs; $(g) - (i)$ illustrate the results of Monte Carlo experiments with 1000 people and 8 hubs. RIIVs: random initially informed vertices; SIIVs: specific initially informed vertices. 5%: the 5% of 1000 initially informed vertices. 10%: the 10% of 1000 initially informed vertices. 15%: the 15% of 1000 initially informed vertices.

figure 4. The results indicate that for short-term spreading with limited time, the SIIVs strategy is capable of significantly improving spreading efficiency in networks with 1000 people.

The improvement of spreading efficiency when using the proposed strategy is increasingly significant with the increase in the number of hubs. The improvement of spreading efficiency when using the proposed strategy is not increasingly significant with the increase in the number of IIVs, and there is a weakly decreased improvement. Moreover, the improvement of spreading efficiency when using the proposed strategy in networks with 1000 people is better than that in networks with 500 people.

## 4.4. Results of the Monte Carlo experiments with 2000 people

In this subsection, results of the Monte Carlo experiments with 2000 people are presented, including the results of Monte Carlo experiments with 2000 people and 8 hubs, 12 hubs and 16 hubs, respectively; see figure 5. The results indicate that for the short-term spreading with limited time, the SIIVs strategy is capable of significantly improving spreading efficiency in networks with 2000 people.

The improvement of spreading efficiency when using the proposed strategy is not increasingly significant with the increase in the number of hubs. The improvement of spreading efficiency when using the proposed strategy is also not increasingly significant with the increase in the number of IIVs, and the improvement seems to be stable with a different number of IIVs or hubs. Moreover, the

rsos.royalsocietypublishing.org    R. Soc. open sci. **5**: 181137

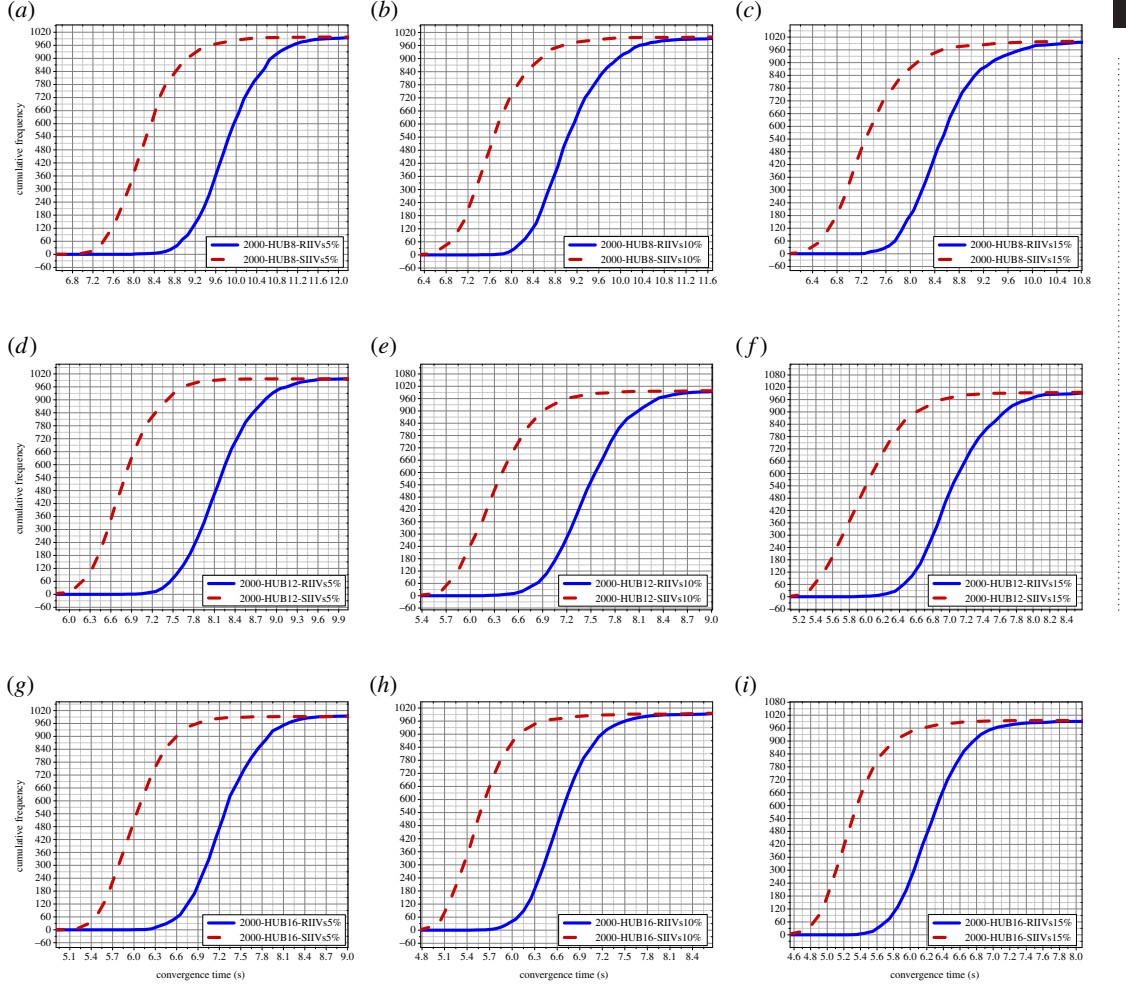

**Figure 5.** Results of the Monte Carlo experiments with 2000 people. $(a)-(c)$ illustrate the results of Monte Carlo experiments with 2000 people and 8 hubs; $(d)-(f)$ illustrate the results of Monte Carlo experiments with 2000 people and 12 hubs; $(g)-(i)$ illustrate the results of Monte Carlo experiments with 2000 people and 16 hubs. RIIVs: Random initially informed vertices; SIIVs: Specific initially informed vertices. 5%: the 5% of 2000 initially informed vertices. 10%: the 10% of 2000 initially informed vertices. 15%: the 15% of 2000 initially informed vertices.

improvement of the spreading efficiency when using the proposed strategy in the networks with 2000 people is better than that in the networks with 500 people and 1000 people.

## 5. Discussion

We discuss the following four issues in this section: (1) the comparison of the RIIVs and the SIIVs regarding spreading efficiency; (2) the effect of different numbers of SIIVs on spreading efficiency; (3) the effect of different numbers of hubs on spreading efficiency and (4) the effect of different network sizes on spreading efficiency.

### 5.1. Comparison of the *RIIVs* and the *SIIVs* in terms of spreading efficiency in different networks

We can directly see spreading efficiency when using the RIIVs and the SIIVs in different networks in figures 3–5. In each figure, the imaginary line expresses spreading when using the SIIVs strategy; the full line expresses spreading when using the RIIVs strategy; the top location of the circled-blue line illustrates that spreading when using the SIIVs is more efficient than spreading when using the RIIVs strategy in short-term spreading with limited time. We verify that the influential initial spreaders can

rsos.royalsocietypublishing.org    R. Soc. open sci. **5**: 181137

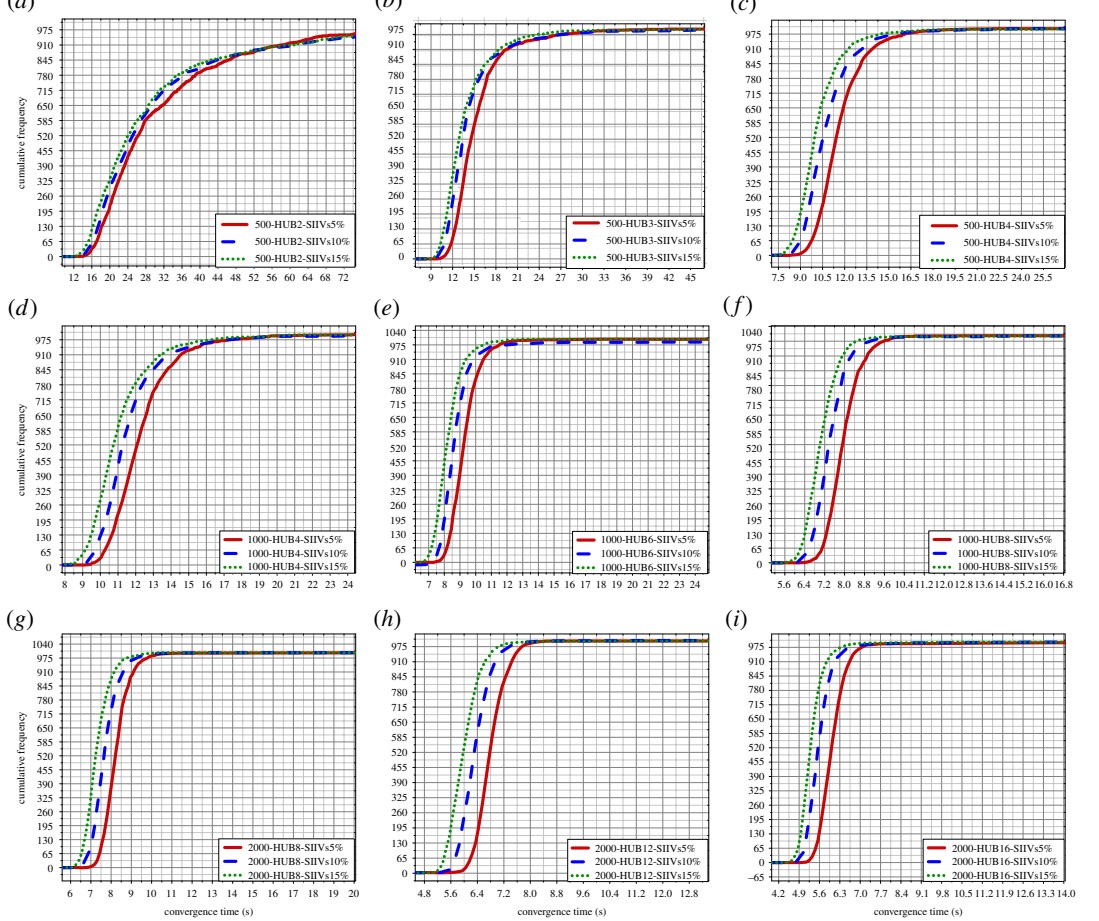

**Figure 6.** Comparison of spreading efficiency when setting different numbers of SIIVs. $(a)-(c)$ illustrate the comparison of spreading efficiency when setting different numbers of SIIVs in the networks with 500 people and 2 hubs, 3 hubs, 4 hubs, respectively; $(d)-(f)$ illustrate the comparison of spreading efficiency when setting different numbers of SIIVs in the networks with 1000 people and 4 hubs, 6 hubs, 8 hubs, respectively; $(g)-(i)$ illustrate the comparison of spreading efficiency when setting different numbers of SIIVs in the networks with 2000 people and 8 hubs, 12 hubs, 16 hubs, respectively. SIIVs: specific initially informed vertices. SIIVs 5%, 10% and 15% express that the number of IIVs is the 5%, 10% and 15% of the network size, respectively.

improve spreading efficiency, and we also demonstrate that a set of large-degree vertices can improve the spreading efficiency significantly when they are the initial spreaders.

Spreading when using two strategies can reach the same performance when there is no limitation for spreading time, and our results demonstrated this. However, for short-term spreading with limited time, spreading needs to be as fast as possible at the beginning. Our results demonstrated that spreading when using the proposed strategy can satisfy this requirement.

## 5.2. Effect of *different numbers of SIIVs* on spreading efficiency in different networks

In this subsection, we compare and evaluate the spreading efficiency when setting different numbers of SIIVs in the network with the same network size and the same number of hubs. In this case, there are nine groups of data; see figure 6.

In figure 6, we find that there are small differences between the efficiency of spreading when setting different numbers of SIIVs. This finding probably is observed because the configured intervals of the numbers of SIIVs are too small, and this configuration for intervals cannot lead to significant differences between spreading efficiency when setting different numbers of SIIVs.

Moreover, we can observe that for the improvement of spreading efficiency when the number of SIIVs is increased, the improvement is more significant in the networks with a large size than in the networks with a small size. That improvement could be caused by the different connections in

rsos.royalsocietypublishing.org    R. Soc. open sci. **5**: 181137

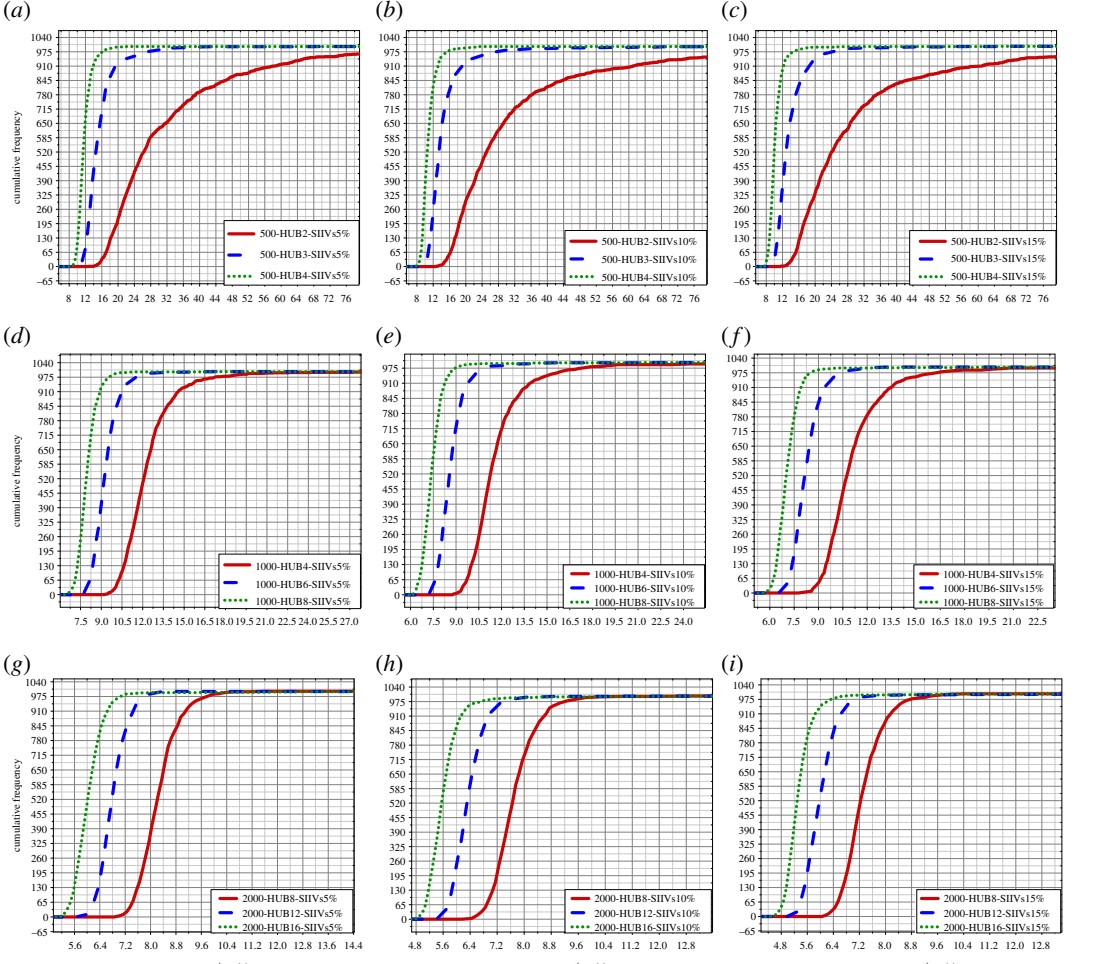

**Figure 7.** Comparison of spreading efficiency in networks with different numbers of hubs. $(a) - (c)$ illustrate the comparison of spreading efficiency when setting the same number of SIIVs in the networks with 500 people and 2 hubs, 3 hubs, 4 hubs; $(d) - (f)$ illustrate the comparison of spreading efficiency when setting the same number of SIIVs in the networks with 1000 people and 4 hubs, 6 hubs, 8 hubs; $(g) - (i)$ illustrate the comparison of spreading efficiency when setting the same number of SIIVs in the networks with 2000 people and 8 hubs, 12 hubs, 16 hubs. SIIVs: Specific initially informed vertices. SIIVs 5%, 10%, and 15% express that the number of IIVs is the 5%, 10% and 15% of the network size, respectively.

different networks. According to our experimental configurations, there are more connections in the networks with a large size because of a large number of hubs. The connections of the initial spreaders are increased with the increase in the number of SIIVs, and the increase is more significant in networks with a large size than in networks with a small size. It is obvious that spreading is more efficient with more initial connections for spreading information.

### 5.2.1. Effect of *different numbers of hubs* on the spreading efficiency in different networks when setting the same number of SIIVs.

In this subsection, we compare spreading efficiency when setting the same number of SIIVs in networks with different settings of the number of hubs and the same setting of network size. There are nine groups of data in this case; see figure 7.

In figure 7, we find that different settings of the number of hubs in the networks can significantly impact spreading efficiency, and spreading is more efficient in the network with the larger number of hubs, for the same setting of the number of SIIVs and the same setting of network size. This is possibly caused because with the same setting of network size and the same setting of the number of SIIVs, the setting of the larger number of hubs in the network indicates that vertices have the more connections in the network, and it is clear that spreading is faster and more efficient when

rsos.royalsocietypublishing.org    R. Soc. open sci. **5**: 181137

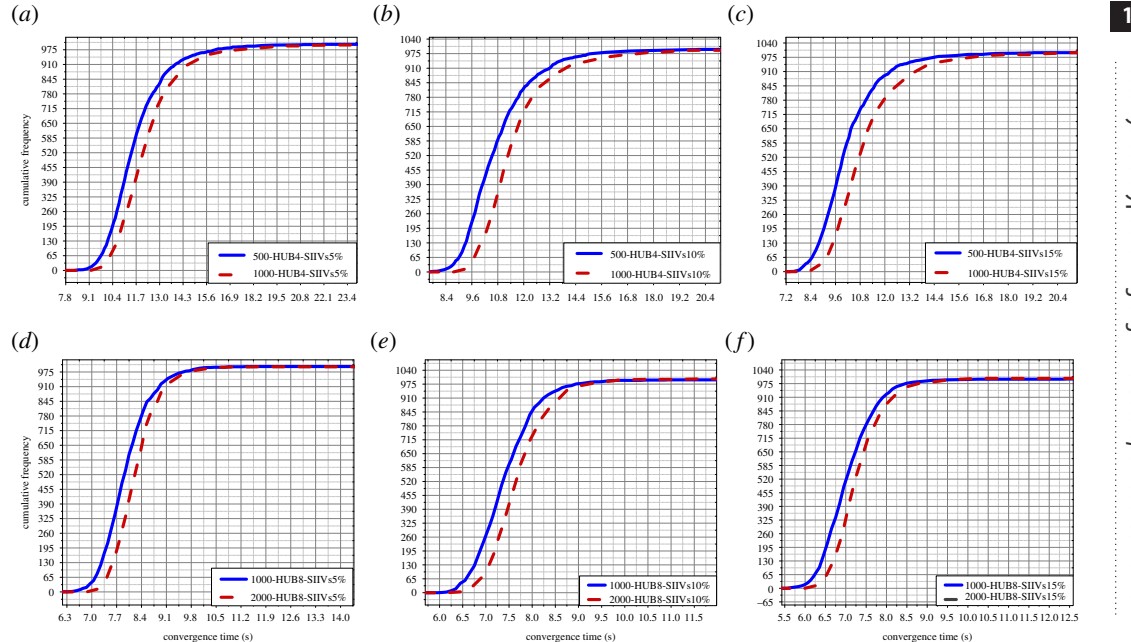

**Figure 8.** Comparison of spreading efficiency in the networks with different sizes. $(a)-(c)$ illustrate the comparison of spreading efficiency when setting the same number of SIIVs and the same number of hubs (i.e. 4 hubs) in the networks with 500 people and 1000 people; $(d)-(f)$ illustrate the comparison of spreading efficiency when setting the same number of SIIVs in the networks with 1000 people and 2000 people. SIIVs: Specific initially informed vertices. SIIVs 5%, 10%, and 15% express that the number of IIVs is the 5%, 10%, and 15% of the network size, respectively.

more connections of the vertices can be employed to spread the information to the same number of uninformed vertices.

Moreover, we also find that the improvement of the spreading efficiency gradually decreases with the increase in the number of hubs in the networks with the same size and the same number of SIIVs. We speculate that the improvement of spreading efficiency could be convergent when the number of hubs is large enough in the networks with the same size and the same number of SIIVs.

## 5.3. Effect of *different sizes* on spreading efficiency in different networks when setting the same number of SIIVs

In this subsection, we compare the spreading efficiency when setting the same number of SIIVs in the networks with different sizes and the same number of hubs. There are six groups of data; see figure 8.

In figure 8, we can find that spreading is more efficient when the network size is smaller with the same setting of the number of SIIVs and the same setting of the number of hubs. This finding is probably attributable to the following reason. In the scale-free networks, the same setting of the number of hubs indicates that the same number of vertices have many connections in the networks. The same setting of the number of SIIVs indicates that the same number of IIVs know the information at the beginning of spreading. The setting of the smaller network size indicates that a smaller number of vertices needs to be informed in totality. For the same setting of the number of SIIVs and the same setting of the number of hubs, the spreading is clearly more rapid and efficient when the network size is smaller.

## 6. Conclusion

In this paper, we proposed a selection strategy for improving short-term spreading efficiency in scale-free networks by specifying a set of top large-degree vertices as the initially informed vertices. The essential idea behind the proposed method is (1) to select the initial spreaders according to degree centrality with low complexity and (2) to exploit the significant diffusion of the top large-degree vertices at the beginning of spreading. We first built a multi-agent information spreading model based on 6

assumptions to evaluate the performance of our proposed method. We then conducted 54 Monte Carlo experiments with 9 scale-free network structures and 3 numbers of initially informed vertices. The results indicated the following: (1) short-term spreading efficiency could be improved when a set of top large-degree vertices were selected as the initial spreaders; (2) the improvement of spreading efficiency was more significant in networks with a large size than in networks with a small size; (3) for the same setting of the network size and the same setting of the number of SIIVs, spreading is more efficient in the network with the larger number of hubs, and the improvement of spreading efficiency gradually decreases with the increase in the number of hubs; (4) for the same setting of the number of initially informed vertices and the same setting of the number of the hubs, the spreading is more efficient in the smaller network. This proposed selection strategy could be applied in warning or crisis information spreading. In the future, we will evaluate the impact of network structures on spreading efficiency.

Data accessibility. Our data have been deposited at Dryad (https://doi.org/10.5061/dryad.8mj8fc5). [38]
Authors' contributions. S. Y. W. and Y. F. D devised the research project. S. Y. W. and Y. L. built the relevant models and performed numerical simulations. S. Y. W., Y. F. D. and Y. L. analysed the results and wrote the paper.
Competing interests. We declare we have no competing interests.
Funding. This work was supported by the National Key Research Project of China(grant no. 2017YFC0804706) and the National Natural Science Foundation of China(grant nos. 91324017, 71642005, 71774098.
Acknowledgements. We acknowledge the useful insights of the anonymous referees.

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
