## [Reviewer comments · Royal Society Open Science]

Review History

RSOS-180404.R0 (Original submission)

Review form: Reviewer 1 (Salvatore Cuomo)

Is the manuscript scientifically sound in its present form?

Yes

Are the interpretations and conclusions justified by the results?

Yes

Is the language acceptable?

Yes

Is it clear how to access all supporting data?

Yes

Do you have any ethical concerns with this paper?

No

Have you any concerns about statistical analyses in this paper?

Yes

Recommendation?

Accept with minor revision (please list in comments)

Comments to the Author(s)

The paper describes a selection strategy for spreading information by specifying a set of top large-degree vertices. The main idea is to exploit the significant diffusion of the top large-degree vertices at the beginning of spreading. The approach seems to be novel and very interesting. I suggest to accept this paper after minor revisions.

i) Please discuss this approach more in general in the introduction section

ii) Please explain better fig. 2

iii) Discuss better the Monte Carlo Method.

iv) detail better the experimental section

Review form: Reviewer 2

Is the manuscript scientifically sound in its present form?

Yes

Are the interpretations and conclusions justified by the results?

Yes

Is the language acceptable?

Yes

Is it clear how to access all supporting data?

No

Do you have any ethical concerns with this paper?

No

Have you any concerns about statistical analyses in this paper?

No

Recommendation?

Reject

Comments to the Author(s)

The authors describe the a selection strategy for spreading information called SIIV, which consists in properly detecting a set of top-degree vertices as the Initial Informed Vertices.

The paper is well written and the overall organization is good. However, is not completely proof-checked (you need to correct several typos in it) and there are some problems in tables 2

and 3 (e.g., the specification of RIIV and SIIV). In addition, the figures are not clear since you shouldn't choose colors referring to lines, but, e.g., different line styles.

The paper is a good ad a "guideline paper" in describing the topic of information spreading in networks, but it is not particularly clear to me the contribution you provided and the novelty of the paper. Despite the Information Spreading Model that you propose is quite reasonable, the experiments you design are not useful. In fact, e.g., it is obvious that a proper choice of initial Informed Vertices performs better than a random one, and this is proved by the fact that, after a transient, both strategies reach the same performance. Moreover, it is not clear how you choose the "specific" IIV.

Decision letter (RSOS-180404.R0)

22-May-2018

Dear Dr Wang:

Manuscript ID RSOS-180404 entitled "An Efficient Method for Short-term Information Spreading in Scale-free Networks by Specifying the Initially Informed Vertices" which you submitted to Royal Society Open Science, has been reviewed. The comments from reviewers are included at the bottom of this letter.

In view of the criticisms of the reviewers, the manuscript has been rejected in its current form. However, a new manuscript may be submitted which takes into consideration these comments.

Please note that resubmitting your manuscript does not guarantee eventual acceptance, and that your resubmission will be subject to peer review before a decision is made.

Your resubmitted manuscript should be submitted by 19-Nov-2018. If you are unable to submit by this date please contact the Editorial Office.

Please note that Royal Society Open Science will introduce article processing charges for all new submissions received from 1 January 2018. Charges will also apply to papers transferred to Royal Society Open Science from other Royal Society Publishing journals, as well as papers submitted as part of our collaboration with the Royal Society of Chemistry (<http://rsos.royalsocietypublishing.org/chemistry>). If your manuscript is submitted and accepted for publication after 1 Jan 2018, you will be asked to pay the article processing charge, unless you request a waiver and this is approved by Royal Society Publishing. You can find out more about the charges at <http://rsos.royalsocietypublishing.org/page/charges>. Should you have any queries, please contact openscience@royalsociety.org.

Kind regards,
Andrew Dunn
Senior Publishing Editor
Royal Society Open Science
openscience@royalsociety.org

on behalf of Prof Marta Kwiatkowska (Subject Editor)
openscience@royalsociety.org

Reviewers' Comments to Author:

Reviewer: 1

Comments to the Author(s)

The paper describes a selection strategy for spreading information by specifying a set of top large-degree vertices. The main idea is to exploit the significant diffusion of the top large-degree vertices at the beginning of spreading. The approach seems to be novel and very interesting. I suggest to accept this paper after minor revisions.

i) Please discuss this approach more in general in the introduction section

ii) Please explain better fig. 2

iii) Discuss better the Monte Carlo Method.

iv) detail better the experimental section

Reviewer: 2

Comments to the Author(s)

The authors describe the a selection strategy for spreading information called SIIV, which consists in properly detecting a set of top-degree vertices as the Initial Informed Vertices. The paper is well written and the overall organization is good. However, is not completely proof-checked (you need to correct several typos in it) and there are some problems in tables 2 and 3 (e.g., the specification of RIIV and SIIV). In addition, the figures are not clear since you shouldn't choose colors referring to lines, but, e.g., different line styles.

The paper is a good ad a "guideline paper" in describing the topic of information spreading in networks, but it is not particularly clear to me the contribution you provided and the novelty of the paper. Despite the Information Spreading Model that you propose is quite reasonable, the experiments you design are not useful. In fact, e.g., it is obvious that a proper choice of initial Informed Vertices performs better than a random one, and this is proved by the fact that, after a transient, both strategies reach the same performance. Moreover, it is not clear how you choose the "specific" IIV.

Author's Response to Decision Letter for (RSOS-180404.R0)

See Appendices A - C.

RSOS-181137.R0

Review form: Reviewer 2

Is the manuscript scientifically sound in its present form?

Yes

Are the interpretations and conclusions justified by the results?

Yes

Is the language acceptable?

Yes

Is it clear how to access all supporting data?

Yes

Do you have any ethical concerns with this paper?

No

Have you any concerns about statistical analyses in this paper?

No

Recommendation?

Accept as is

Comments to the Author(s)

The authors completely reply to my questions and fixed the issues

Decision letter (RSOS-181137.R0)

01-Oct-2018

Dear Dr Wang,

I am pleased to inform you that your manuscript entitled "Improving Short-Term Information Spreading Efficiency in Scale-Free Networks by Specifying Top Large-Degree Vertices as the Initial Spreaders" is now accepted for publication in Royal Society Open Science.

Royal Society Open Science operates under a continuous publication model (<http://bit.ly/cpFAQ>). Your article will be published straight into the next open issue and this will be the final version of the paper. As such, it can be cited immediately by other researchers.

As the issue version of your paper will be the only version to be published I would advise you to check your proofs thoroughly as changes cannot be made once the paper is published.

You have the opportunity to archive your accepted, unbranded manuscript, but access to the full text must be embargoed until publication.

Articles are normally press released. For this to be effective we set an embargo on news coverage corresponding to the publication date of the article. We request that news media and the authors do not publish stories ahead of this embargo (when final version of the article is available).

on behalf of Prof Marta Kwiatkowska (Subject Editor)
openscience@royalsociety.org

Reviewer comments to Author:

Reviewer: 2

Comments to the Author(s)

The authors completely reply to my questions and fixed the issues

Appendix A

List of Modifications in the Revised Manuscript

Shuangyan Wang¹, Yunfeng Deng^{2*}, Ying Li¹

1: School of Engineering and Technology, China University of Geosciences
Beijing

2: Chinese Academy of Governance

sy.wang@cugb.edu.cn; 13910185162@139.com; liying@cugb.edu.cn

Acknowledgement The authors are grateful to the editor and the anonymous reviewers for a careful checking of the details and for helpful comments that improved this paper.

Modification of Main Text

1. There is a minor revision in the **Section of Abstract**.
2. The third and fourth paragraphs in **Section of Introduction** are revised.
3. The fourth paragraphs in **Section of Problem Statement** is revised.
4. The **Subsection of Design of Comparative Experiments** is revised.
5. The **Section of Results** is revised.
6. The **Section of Discussion** is revised.
7. The **Section of Conclusion** is revised.

Modification of Tables

1. Some descriptions in **Table 1~3** are revised.
2. The descriptions of RIIV and SIIV are added in **Table 2~3**.

Modification of Figures

1. Some descriptions in **Figure 2** are revised.
2. The line styles in **Figure 3~8** are revised.

Modification of References

The following references are newly added:

20. Cuomo, S., et al., A Biologically Inspired Model for Analyzing Behaviours in Social Network Community and Cultural Heritage Scenario. 10th International Conference on Signal-Image Technology and Internet-Based Systems Sitis 2014, ed. K. Yetongnon, A. Dipanda, and R. Chbeir. 2014. 485-492.
21. Cuomo, S., et al., A Cultural Heritage case study of visitor experiences shared on a Social Network. 2015 10th International Conference on P2p, Parallel, Grid, Cloud and Internet Computing, ed. F. Xhafa, et al. 2015. 539-544.
29. Anshelevich, E., A. Hate, and M. Magdon-Ismail, Seeding influential nodes in non-submodular models of information diffusion. Autonomous Agents and Multi-Agent Systems, 2014. 29(1): p. 131-159.
30. Guo, L., et al., Identifying multiple influential spreaders in term of the distance-based coloring. Physics Letters, Section A: General, Atomic and Solid State Physics, 2016. 380(7-8): p. 837-842.
31. Fu, Y.H., C.Y. Huang, and C.T. Sun, Using global diversity and local topology features to identify influential network spreaders. Physica A: Statistical Mechanics and its Applications, 2015. 433: p. 344-355.
33. Anylogic. Official Site. 2018; Available from: <https://www.anylogic.com/>.
34. Barabasi, A.L. and E. Bonabeau, Scale-free networks. Scientific American, 2003. 288(5): p. 60-69.
37. Wikipedia. Monte Carlo Method. 2018; Available from: https://en.wikipedia.org/wiki/Monte_Carlo_method.

The authors are grateful to the editor and anonymous reviewers for helpful comments!

Appendix B

Responses to the First Reviewer's Comments

Shuangyan Wang¹, Yunfeng Deng^{2*}, Ying Li¹

1: School of Engineering and Technology, China University of Geosciences
Beijing

2: Chinese Academy of Governance

sy.wang@cugb.edu.cn; 13910185162@139.com; liying@cugb.edu.cn

Acknowledgement The authors are grateful to the editor and the anonymous reviewers for a careful checking of the details and for helpful comments that improved this paper.

Please note that the modifications made in the revised manuscript are listed on a document named as “**List of Modifications. pdf**”.

Comment # 1:

Please discuss this approach more in general in the introduction section.

Response:

First, thank you so much for giving this very critical and valuable comment!

We have revised this part in the manuscript, and the revised part can be seen in **Section 1 (Introduction)**.

Comment # 2:

Please explain better fig. 2.

Response:

We have revised this part in the manuscript, and the revised part can be seen in **Section 3.2 (Design of Comparative Experiments)**.

Comment # 3:

Discuss better the Monte Carlo Method.

Response:

We have revised this part in the manuscript, and the revised part can be seen in **Section 3.2 (Design of Comparative Experiments)**.

Comment # 4:

Details better the experimental section.

Response:

We have revised this part in the manuscript, and the revised part can be seen in **Section 3.2 (Design of Comparative Experiments)**.

The authors are grateful to the anonymous first reviewer for helpful comments!

Appendix C

Responses to the Second Reviewer's Comments

Shuangyan Wang¹, Yunfeng Deng^{2*}, Ying Li¹

1: School of Engineering and Technology, China University of Geosciences
Beijing

2: Chinese Academy of Governance

sy.wang@cugb.edu.cn; 13910185162@139.com; liying@cugb.edu.cn

Acknowledgement The authors are grateful to the editor and the anonymous reviewers for a careful checking of the details and for helpful comments that improved this paper.

Please note that the modifications made in the revised manuscript are listed on a document named as “**List of Modifications. pdf**”.

Comment # 1:

The paper is well written and the overall organization is good. However, is not completely proof-checked (you need to correct several typos in it)

Response:

First, thank you so much for giving this very critical and valuable comment!

We have checked our manuscript carefully.

Comment # 2:

There are some problems in tables 2 and 3 (e.g., the specification of RIIV and SIIV).

Response:

We have added the descriptions of the RIIV and SIIV in Table 2 and 3, and we have checked the two tables carefully.

Comment # 3:

In addition, the figures are not clear since you shouldn't choose colors referring to lines, but, e.g., different line styles.

Response:

We have revised all figures with different line styles in the manuscript.

Comment # 4:

The paper is a good ad a "guideline paper" in describing the topic of information spreading in networks, but it is not particularly clear to me the contribution you provided and the novelty of the paper. Despite the Information Spreading Model that you propose is quite reasonable, the experiments you design are not useful. In fact, e.g., it is obvious that a proper choice of initial Informed Vertices performs better than a random one, and this is proved by the fact that, after a transient, both strategies reach the same performance.

Response:

We agree that a proper choice of initial informed vertices performs better than a random one, and our experimental results have proved this point. However, it is also a good question that how to select the initial spreaders. Actually, many metrics and methods are proposed for identifying the initial spreaders, but for the short-term spreading with limited time, the identified method must be of low complexity and low power consumption. In this paper, we select the degree centrality as the metric to measure the influence of initial spreaders, because of the low complexity for identifying the large-degree vertices and the strong diffusion of large-degree vertices. We propose a selection strategy for improving spreading efficiency which is suitable to be applied to the short-term spreading.

Moreover, if there no limitation of the spreading time, it is obviously that all spreading can reach the same performance. However,

in the short-term spreading, the spreading time is limited, the essential question is that how many people can receive the information during a limited time. And in this paper, we terminate experiments until the spreading is convergent because that we want to record the completely experimental process. In real situations, the spreading cannot be convergent because of the limited spreading time. For the short-term spreading, it is important to improve the spreading efficiency at the beginning of the spreading or during a short limited time.

Comment # 5:

Moreover, it is not clear how you choose the "specific" IIV.

Response:

We select the initial informed spreader according to the degree centrality of vertices. The specific procedure is that: we first sort all vertices according to the degree centrality on descending order. Then we select a set of top large-degree vertices as the initial informed vertices. This method is of low computational complexity, and the degree centrality can measure the strong diffusion of vertices. The degree centrality is suitable to be employed in the short-term spreading. In the real emergency training, it is easy to find the large-degree persons in a community, they are always the leaders, managers, or organizers.

The authors are grateful to the anonymous second reviewer for helpful comments!